

# Sediment microbial taxonomic and functional diversity in a natural salinity gradient challenge Remane's "species minimum" concept

Christina Pavloudi[1,2,3], Jon B. Kristoffersen[1], Anastasis Oulas[1,4], Marleen De Troch[2] and Christos Arvanitidis[1]

[1] Institute of Marine Biology, Biotechnology and Aquaculture (IMBBC), Hellenic Centre for Marine Research (HCMR), Heraklion, Crete, Greece
[2] Marine Biology Research Group, Department of Biology, Faculty of Sciences, Ghent University, Ghent, Belgium
[3] Microbial Ecophysiology Group, Faculty of Biology/Chemistry and MARUM, University of Bremen, Bremen, Germany
[4] Bioinformatics Group, The Cyprus Institute of Neurology and Genetics, Nicosia, Cyprus

Corresponding author
Christina Pavloudi,
cpavloud@hcmr.gr

## ABSTRACT

Several models have been developed for the description of diversity in estuaries and other brackish habitats, with the most recognized being Remane's Artenminimum ("species minimum") concept. It was developed for the Baltic Sea, one of the world's largest semi-enclosed brackish water body with a unique permanent salinity gradient, and it argues that taxonomic diversity of macrobenthic organisms is lowest within the horohalinicum (5 to 8 psu). The aim of the present study was to investigate the relationship between salinity and sediment microbial diversity at a freshwater-marine transect in Amvrakikos Gulf (Ionian Sea, Western Greece) and assess whether species composition and community function follow a generalized concept such as Remane's. DNA was extracted from sediment samples from six stations along the aforementioned transect and sequenced for the 16S rRNA gene using high-throughput sequencing. The metabolic functions of the OTUs were predicted and the most abundant metabolic pathways were extracted. Key abiotic variables, i.e., salinity, temperature, chlorophyll-a and oxygen concentration etc., were measured and their relation with diversity and functional patterns was explored. Microbial communities were found to differ in the three habitats examined (river, lagoon and sea) with certain taxonomic groups being more abundant in the freshwater and less in the marine environment, and *vice versa*. Salinity was the environmental factor with the highest correlation to the microbial community pattern, while oxygen concentration was highly correlated to the metabolic functional pattern. The total number of OTUs showed a negative relationship with increasing salinity, thus the sediment microbial OTUs in this study area do not follow Remane's concept.

## INTRODUCTION

Salinity is considered as one the most influential environmental factors, not only for the distribution of benthic and pelagic organisms (*Remane, 1934*; *Rolston & Dittmann, 2009*; *Palmer et al., 2011*; *Darr, Gogina & Zettler, 2014*), but also for microbial community composition (e.g., *Barcina, Lebaron & Vives-Rego, 2006*; *Wu et al., 2006*; *Lozupone & Knight, 2007*; *Logares et al., 2009*). Bacterial abundance, activity and growth can be affected by salinity (*Ben-Dov, Brenner & Kushmaro, 2007*; *Caporaso et al., 2011*) and, in certain cases salinity can induce mortality of bacteria, thus regulating bacterial abundance in some estuaries (*Painchaud, Therriault & Legendre, 1995*). Specifically, salinity fluctuations, and their subsequent effect on aquatic biota, are more noticeable in estuaries and other brackish water bodies, as these habitats are characterized by a more or less pronounced salinity gradient (*Telesh, Schubert & Skarlato, 2013*).

It has been suggested that in brackish water ecosystems, taxonomic diversity of macrobenthic organisms is lowest within the horohalinicum, which occurs at salinity 5 to 8 psu, because the number of brackish specialist species does not compensate for the decline of the marine and freshwater species richness (*Remane, 1934*). This concept, referred to as the Remane's Artenminimum ('species minimum') concept originated from the Baltic Sea, one of the world's largest semi-enclosed, brackish water body with a unique permanent salinity gradient (*Telesh, Schubert & Skarlato, 2011*). Despite being developed for the Baltic Sea, Remane's concept became the recognized model for the description of diversity in estuaries and other brackish habitats (*McLusky & Elliott, 2004*). However, alternative models challenging Remane's concept have also been developed. In certain cases a reverse curve has been observed, with the peak of species occurring in the horohalinicum (*Telesh, Schubert & Skarlato, 2011*) while in others a linear decrease (*Attrill, 2002*) or even no change (*Herlemann et al., 2011*) in the number of species across the salinity gradient were observed.

Remane's concept can be projected in other aquatic bodies with similar evolution to the Baltic Sea, such as the Amvrakikos Gulf (Ionian Sea, Western Greece) (*Ferentinos et al., 2010*). The Gulf was formed in the Middle Quaternary period (*Anastasakis, Piper & Tziavos, 2007*); the marine transgression took place at approximately 11 ka BP and the Gulf attained its present shape at approximately 4 ka BP (*Kapsimalis et al., 2005*). In addition, the low tidal range (on average 5 cm) and the low energy wave regime prevailing in Amvrakikos Gulf (*Ferentinos et al., 2010*), render the latter as a Baltic Sea analogue in the Mediterranean Sea.

In the light of projected climate changes and the subsequent sea level rise and saltwater intrusion that will occur, microbial populations in freshwater wetlands near the coast will be subjected to elevated salinities (*Chambers, Reddy & Osborne, 2011*). Due to the long-term effect of sea level rise, saltwater intrusion can affect ecosystems on timescales of decades (*Neubauer, Franklin & Berrier, 2013*). It is therefore crucial to explore the current status of microbial communities in wetlands in order to comprehend the impact of such acute changes in their diversity patterns, since they are involved in biogeochemical processes that are crucial for maintaining the planet in a habitable state (e.g., *Falkowski, Fenchel & Delong, 2008*).

Currently, state-of-the-art studies of microbial communities through high throughput sequencing of the 16S rRNA gene, i.e., marker gene metagenomics (*Oulas et al., 2015*), allow documentation of the high diversity of microbial communities in a variety of habitats, e.g., wetlands (*Wang et al., 2012*; *Jiang et al., 2013*), estuaries (*Bobrova et al., 2016*), lakes (*Zhang et al., 2015*), rivers (*Staley et al., 2013*) and coastal lagoons (*Highton et al., 2016*). The substantial data derived from such techniques can be used to test ecological hypotheses, on which sound conclusions can be drawn; for example if microorganisms have a biogeography (*Zinger et al., 2011*) or if their biodiversity is driven by changes in elevation (*Fierer et al., 2011*).

The aim of the present study was to investigate the sediment prokaryotic diversity along a transect river-lagoon-open sea, i.e., from freshwater to marine, in Amvrakikos Gulf in order to test whether it follows a generalized concept, such as Remane's, both in terms of Operational Taxonomic Unit (OTU) composition as well as on distribution of metabolic functions. If the applicability of the concept is confirmed in Amvrakikos Gulf, it would enhance its transferability to other brackish water bodies than the Baltic Sea. In addition, confirmation of Remane's concept for prokaryotes would mean that small-bodied, fast-developing and rapidly evolving microbes respond in a similar way as benthic macroorganisms in a salinity gradient.

## MATERIALS & METHODS

### Field sampling

The transect river-lagoon-open sea, i.e., from freshwater to marine, chosen for the present study is naturally occurring at Amvrakikos Gulf (Ionian Sea, Western Greece) (Fig. 1). The Gulf is a semi-enclosed embayment connected with the Ionian Sea via a narrow channel, the Preveza (Aktio) Strait (*Kapsimalis et al., 2005*) and is characterized by a fjord-like oceanographic regime (*Ferentinos et al., 2010*). The wetlands of Amvrakikos Gulf are listed in both the Ramsar international convention and the Natura 2000 Network.

The northwest part of the Amvrakikos Gulf is formed by the rivers Arachthos and Louros (*Poulos, Collins & Ke, 1993*; *Poulos, Collins & Lykousis, 1995*; *Vasileiadou et al., 2012*). Arachthos river is the main source of freshwater from November to April (*Poulos, Collins & Ke, 1993*) and its flow is controlled by two hydroelectric dams: the first being an earthfill dam, of 107 m height, mainly built for flood control and the second being a concrete dam, of 42 m height, primarily having a regulatory role, that is to ensure the constant flow of water throughout the year.

Two stations were chosen at the Arachthos river, thus representing the freshwater conditions: one being close to its mouth (39.029690N, 21.025880E) and one in the upper limit of saltwater intrusion when the dams are closed, closed to the village of Neochori (39.070770N, 21.025060E). One more station was chosen at the Arachthos delta (39.039199N, 21.094200E), as an intermediate between the freshwater and the marine realm. In addition, two stations were chosen in the Logarou lagoon due to its vicinity to Arachthos, representing the brackish stations: one at the inner part of the lagoon (39.045528N, 20.902283E), closer to the terrestrial end, and one near the

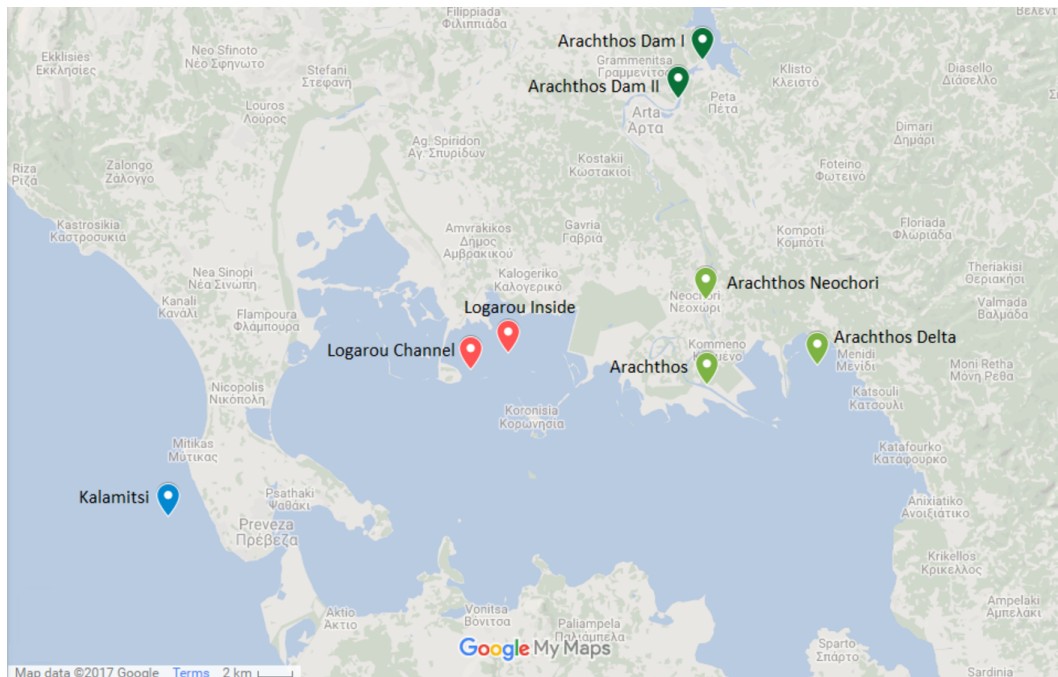

**Figure 1** **Map showing the location of the sampling stations.** Map showing the location of the six sampling stations and the location of the river dams (Map data: ©2017 Google).

channel connecting the lagoon to the Gulf (39.037458N, 20.878626E). Finally, Kalamitsi station (38.966250N, 20.690783E) was chosen from the marine realm, i.e., outside of the Amvrakikos Gulf.

Sampling was carried out in winter of 2014 (November–December), as described in *Pavloudi et al. (2016)*. For the Logarou lagoon and Arachthos river stations, salinity, water temperature and dissolved oxygen concentration were measured in the water overlaying the sediments by means of a portable multi-parameter (WTW Multi 3420 SET G). For the Arachthos delta and Kalamitsi station, water abiotic variables (fluorescence, salinity, water temperature and dissolved oxygen concentration) were recorded with a Sea-Bird Electronics 25 CTD probe. Fluorescence was regarded as a proxy for chlorophyll-a concentration.

Sediment samples from the Logarou lagoon and the Arachthos river were collected by means of a modified manually-operated box-corer, with a sampling surface of 156.25 square centimeters and sediment penetration depth of 25 centimeters, deployed from fishing boats specifically used at the lagoons. Samples from the Arachthos delta and Kalamitsi stations were collected with a Smith McIntyre Grab operated from the R/V Philia. The permission to conduct the field study was provided by the Amvrakikos Wetlands Management Body.

Cylindrical sampling corers (internal sampling surface 15.90 square centimeters) were placed inside the box-corer and the Smith McIntyre Grab in order to collect sub-samples of the sediment's upper layer (0–2 cm). Three replicate units, each retrieved from a different box-corer to avoid pseudoreplication, were taken for each analysis from each sampling station, in order to determine variability within and between stations. Samples for molecular analysis (about 15 cm$^3$ each), i.e., DNA extractions, were placed in 50 ml

falcon tubes (Sarstedt, Nümbrecht, Germany) and were stored at −20 °C, until further processing in the laboratory.

Samples were also collected from cylindrical corers for the measurement of the Particulate Organic Carbon (POC), chloroplast pigments concentration (chlorophyll-a and phaeopigments) and sediment granulometry (for the latter, the sampling depth was four centimeters to allow comparison with previously published data on the study area). To quantify chloroplast pigments concentration in the water column, three replicate water samples were collected from the Logarou lagoon and the Arachthos river by means of Niskin bottles (5 lt). The aforementioned samples were processed at the Chemistry Lab of the IMBBC (HCMR), based on standard techniques (chloroplast pigments: *Yentsch & Menzel, 1963*; POC: *Hedges & Stern, 1984*; granulometry: *Gray & Elliott, 2009*).

## DNA extraction, PCR amplification and 16S rRNA sequencing

DNA was extracted using the PowerSoil DNA Isolation Kit (MO BIO Laboratories, Inc., Carlsbad, CA, USA), as recommended by the manufacturer. About 0.5 (±0.2) grams of wet sediment were used from each sample and the quality of the extracted DNA was evaluated by gel electrophoresis.

PCR amplification was performed targeting the V3–V4 region of the 16S rRNA gene using the bacterial primer pair S-DBact-0341-b-S-17 (or 341F) and S-D-Bact-0785-a-A-21-B (or 805RB), which has been referred to as the most promising primer pair (*Herlemann et al., 2011*; *Klindworth et al., 2013*), with a revision for detection of SAR11 bacterioplankton (*Apprill et al., 2015*).

The Two-Step PCR Approach was used for this study. The first-step PCR was performed with the aforementioned primers containing a universal 5′ tail as specified in the Nextera library protocol from Illumina. The amplification reaction mix of the first PCR contained 6 μl 5x KAPA HiFi Fidelity buffer, 0.9 μl BSA (2 μg/μl), 0.75 μl KAPA dNTP Mix (10 mM), 1.5 μl from each primer (10 μM), 0.75 μl KAPA HiFi HotStart DNA polymerase (1 U/μl) in a final volume of 30 μl per reaction. DNA template concentration was about 10 ng/μl. The first PCR protocol used was the following: 95 °C for 5 min; 25 cycles at 98 °C for 20 s, 57 °C for 2 min, 72 °C for 1 min; 72 °C for 7 min.

The resulting PCR amplicons (∼531 bp) were purified using Agencourt AMPure XP magnetic beads (Beckman Coulter, Brea, CA, USA), quantified using Qubit fluorometric quantitation (Thermo Scientific Fisher, Waltham, MA, USA) and were used as templates for the second-step PCR in order to include the indexes (barcodes), as well as the Illumina adaptors. The amplification reaction mix of the second PCR contained 6 μl 5x KAPA HiFi Fidelity buffer, 0.75 μl KAPA dNTP Mix (10 mM), 3 μl from each primer (10 μM), 0.75 μl KAPA HiFi HotStart DNA polymerase (1 U/μl) in a final volume of 30 μl per reaction. DNA template concentration was about 20 ng/ μl. The second PCR protocol used was the following: 95 °C for 3 min; 8 cycles at 98 °C for 20 s, 55 °C for 30 s, 72 °C for 30 s; 72 °C for 5 min. Amplifications were carried out using T100 Thermal Cycler (BIORAD, Hecules, CA, USA). Again, the resulting PCR amplicons (∼600 bp) were purified and quantified as mentioned previously, mixed in equimolar amounts and sequenced using a MiSeq Reagent Kit v3 (2 × 300-cycles) at the IMBBC (HCMR).

All the raw sequence files of this study were submitted to the European Nucleotide Archive (ENA) (*Leinonen et al., 2011*) with the study accession number PRJEB20211 (available at http://www.ebi.ac.uk/ena/data/view/PRJEB20211).

## Data analyses

The raw sequence reads retrieved from all the sediment samples were quality trimmed using *sickle* (*Joshi & Fass, 2011*), to where the average quality score dropped below 20 (*-q 20*) as well as where read length was below 10 bp (*-l 10*). *SPAdes assembler* (*Bankevich et al., 2012*), which incorporates *BayesHammer* (*Nikolenko, Korobeynikov & Alekseyev, 2013*), was used for the creation of error-corrected paired-end reads, since this strategy along with overlapping paired-end reads reduces errors for MiSeq (*Schirmer et al., 2015*).

Afterwards, *pandaseq* (*Masella et al., 2012*) was used to overlap the paired-end reads using a minimum overlap of 20 (*-o 20*). The overlapped sequences were combined and dereplicated. Then, using *USEARCH* (*Edgar, 2010*), reads were sorted based on abundance, singletons were discarded and OTU clustering and *de novo* chimera removal were performed. Following the relevant recommendation, a reference-based chimera filtering step was performed using *UCHIME* (*Edgar et al., 2011*) using the "Gold" database as a reference.

Reads, including singletons, were then mapped back to OTUs, using the 97% similarity threshold level. Afterwards, they were aligned using *MAFFT* (*Katoh et al., 2005*) and a phylogenetic tree was created using *FastTree* (*Price, Dehal & Arkin, 2010*). Finally, taxonomic profiles of the OTUs were generated using *RDP classifier* (*Wang et al., 2007*).

The metabolic function of the OTUs was predicted using the *Tax4Fun* package (*Aßhauer et al., 2015*), which transforms the SILVA-based OTUs into a taxonomic profile of KEGG organisms (*fctProfiling* $= T$), normalized by the 16S rRNA copy number (*normCopyNo* $= T$). The method for pre-computing the functional reference profiles was the ultrafast protein classification tool (*Meinicke, 2015*) (*refProfile* $= UProC$) and the functional reference profiles were computed based 400 bp reads (*shortReadMode* $= F$). The 100 most abundant metabolic pathways were extracted.

The number of sequences assigned to an OTU represented its relative abundance at a given replicate sample of each sampling station. A matrix containing the microbial OTUs as variables and sampling stations as samples was constructed. A second matrix was constructed, with the predicted metabolic functions of the microbial OTUs as variables and sampling stations as samples. Both matrices were subsequently used to calculate triangular similarity matrices using the Bray–Curtis similarity coefficient (e.g., *Clarke & Warwick, 1994*). In order to test whether the microbial community pattern and the metabolic function pattern could be differentiated based on the sampled habitat, we performed non-metric multidimensional scaling (nMDS) (*Clarke, 1993*) and permutational multivariate analysis of variance (PERMANOVA) (*Anderson, 2001*). The design considered two factors: "habitat" and "location" (999 permutations), with the latter being nested in the former.

A third matrix was also constructed, with the respective abiotic parameters as variables and the sampling stations as samples, which was normalized and used as an input for the BIO-ENV analysis (*Clarke & Ainsworth, 1993*) by employing the Spearman's rank

coefficient. The analysis was performed to identify the subsets of abiotic parameters that were associated with the community (i.e., OTU) and the metabolic function matrices. In order to account for factor confounding, and estimate the amount of variation each factor might explain in the community as well the metabolic function similarity matrices, variation partitioning analysis was performed. The amount of variation in both matrices that was due to salinity uniquely, while taking the other environmental factors into consideration, was estimated and its significance was tested using distance-based redundancy analysis (db-RDA) and Analysis of Variance (ANOVA).

A suite of diversity indices (Margalef's species richness, Pielou's evenness, Shannon-Wiener (*Pielou, 1969*), Chao-1, Abundance Coverage Estimator (ACE)) was calculated for each sampling station. The indices were subsequently tested for significant differences between the different locations and habitats using the Kruskal–Wallis test. The Mann–Whitney *U* test (*Mann & Whitney, 1947*) was used for the *post-hoc* pairwise comparisons; a Bonferroni-correction was applied and the level of significance for the results was lowered to 0.008, in the case of the locations, and to 0.017, in the case of the habitats.

Linear regression was used to examine the significance of relationships between OTU diversity, i.e., the average number of OTUs as well as the average values of diversity indices, and salinity. Shapiro–Wilk test was used to assess normality in the residuals (*Shapiro & Wilk, 1965*), while homoscedastistic residual variances were confirmed by examining plots of the standardized residuals (*Draper & Smith, 1981*).

The non-parametric Kruskal–Wallis test (*Kruskal & Wallis, 1952*) was used to determine which of the predicted metabolic pathways were statistically significant between the different habitats, i.e., "lagoon", "river" and "sea". The *p* value threshold was adjusted from the initial 0.05, using the Benjamini–Hochberg procedure (*Benjamini & Hochberg, 1995*). In addition, the relative abundance values of the main microbial taxa were compared among the locations and habitats of the sampling stations, using the Kruskal–Wallis test.

The *vegan* package (*Oksanen et al., 2008*) was used for the nMDS (*metaMDS* function), PERMANOVA (*adonis* function), BIO-ENV (*bioenv* function), variation partitioning analysis (*varpart* function) and db-RDA (*capscale* function), for the calculation of diversity indices and for the generation of rarefaction curves. Groups in the nMDS plots were displayed using the *ordiellipse* and *veganCovEllipse* functions. Linear regressions, Shapiro–Wilk, Mann–Whitney, Kruskal–Wallis and ANOVA tests were conducted using *stats* package (*R Core Team, 2015*). Graphs were constructed using the *ggplot2* package (*Wickham, 2009*). The aforementioned analyses were performed using R version 3.2.1 (*R Core Team, 2015*).

# RESULTS

## Microbial community composition

The results of the processing of the sequences are shown in Table S1. The 1,893,500 overlapped reads were dereplicated (1,578,523 remained) and singletons were removed (111,223 remained). After *de novo* chimera removal (7,088 OTUs) and chimera removal using the "Gold" database as a reference, the final number of OTUs was 7,050. The corresponding rarefaction curve is shown in Fig. S1.
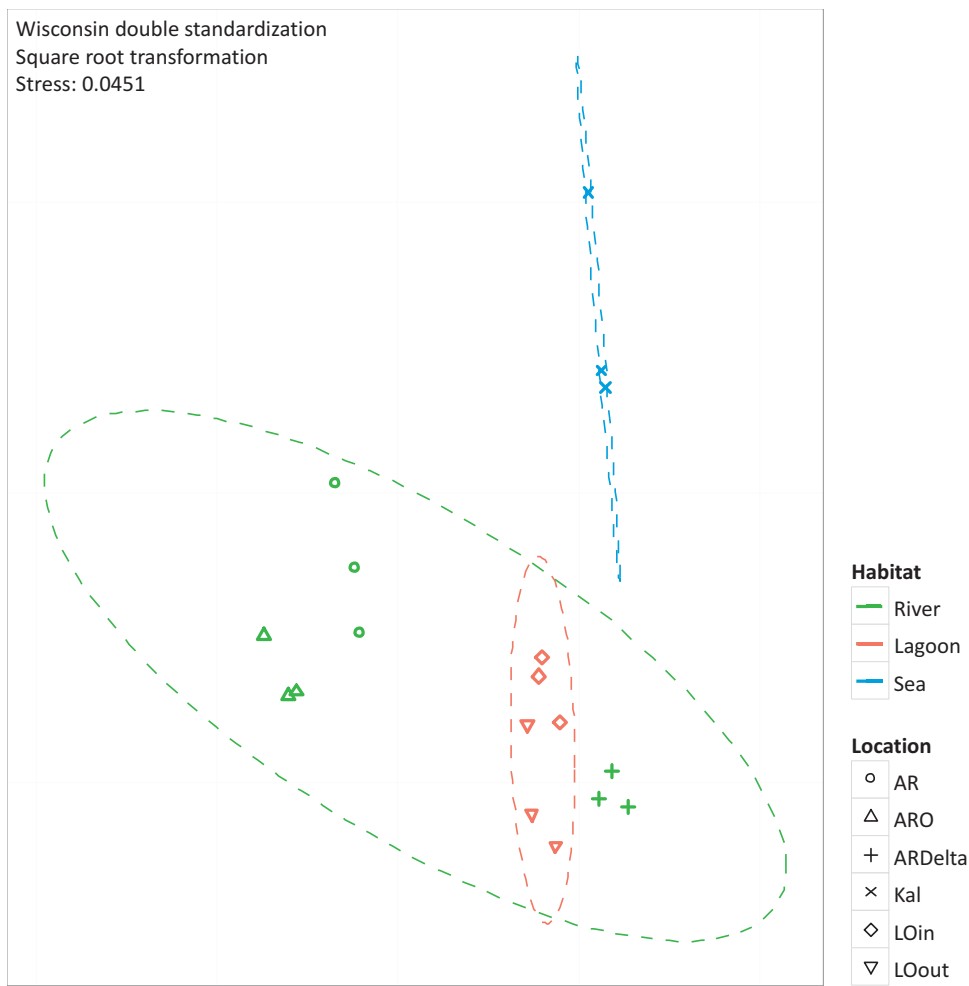

**Figure 2** **nMDS of the microbial OTUs.** nMDS of the similarity matrix of the sampling stations based on the microbial OTUs abundances. Ellipses according to habitat, signs according to location. AR, Arachthos; ARO, Arachthos Neochori; ARDelta, Arachthos Delta; LOin, Logarou station inside the lagoon; LOout, Logarou station in the channel connecting the lagoon to the gulf; Kal, Kalamitsi.

The nMDS of the microbial OTUs (Fig. 2) showed that their spatial pattern differs both by habitat and location, which was also confirmed by the PERMANOVA results (habitat: F.Model $= 19.416$, $p < 0.01$; location: F.Model $= 10.647$, $p < 0.01$).

The relative abundance percentages of the microbial taxa, at the phylum level, did not show a significant differentiation between the different locations or habitats (Kruskal–Wallis: $p > 0.05$ for all cases). However, as shown in Fig. 3 where the relative abundance percentages of each replicate sample have been averaged per sampling station, there are certain differences that can be observed. For example, the Bacteroidetes showed an increasing abundance when moving from the inner station of the river ($\sim$9%) to the more brackish stations ($\sim$28%), and decrease again in the marine station ($\sim$11%). A similar trend was observed for the Proteobacteria, but in this case the higher abundance was found in the marine environment ($\sim$49%). The abundance of Archaea was quite low in all the

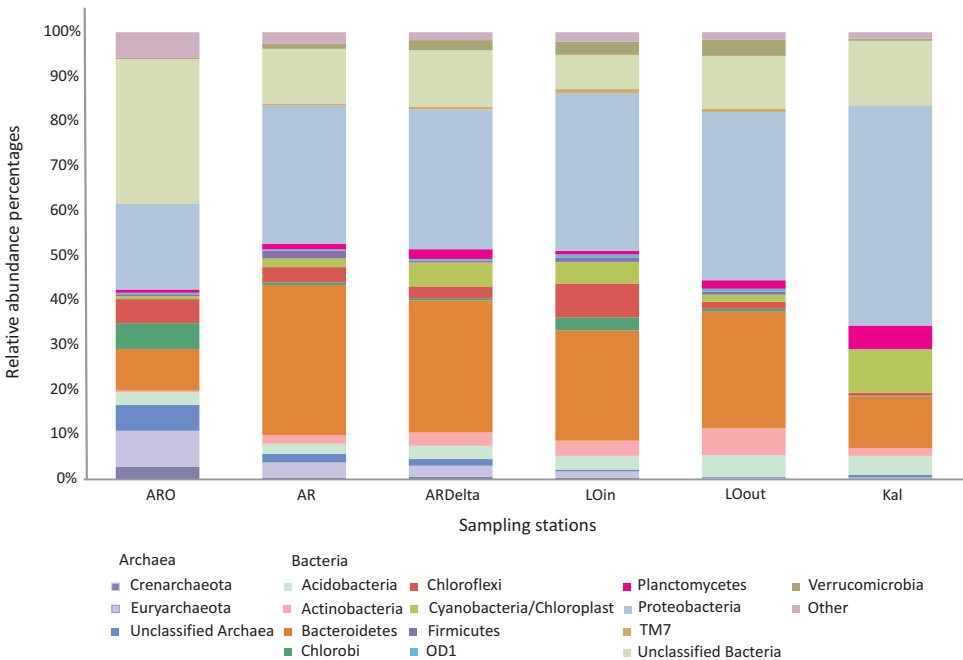

**Figure 3  Bar chart showing the abundances of the main microbial taxa, at the phylum level, at the sampling stations.** AR, Arachthos; ARO, Arachthos Neochori; ARDelta, Arachthos Delta; LOin, Logarou station inside the lagoon; LOout, Logarou station in the channel connecting the lagoon to the gulf; Kal, Kalamitsi.

sampling stations; higher values were observed at the inner station (∼17%) of the river and decrease towards the marine station. Cyanobacteria/Chloroplasts were more abundant in the marine station (∼10%) and were also found in the Arachthos delta and inner station of Logarou lagoon at lower abundance (∼5%).

When the most abundant phyla are examined in more detail, again using the relative abundance percentages of each replicate sample averaged per sampling station, there are certain differences observed at the sampling stations, although non-significant (Kruskal–Wallis: $p > 0.05$ for all the cases). In the case of Bacteroidetes (Fig. S2), Flavobacteria were less abundant at the inner station of the river (∼17%); their abundance was higher in the other riverine stations (∼37%) and in the lagoonal stations (∼54%) while remaining stable at the marine station (∼49%). Sphingobacteria exhibited the lower abundance in the Arachthos delta (∼7%) and the highest in Kalamitsi (∼28%).

Regarding the Proteobacteria phylum (Fig. S3), Alphaproteobacteria exhibited higher abundances at the mouth of the Arachthos river (∼49%) and at the marine station (∼56%). Betaproteobacteria were almost exclusively present in the inner station and mouth of the Arachthos river (∼30% and ∼11% respectively). Gammaproteobacteria were mostly abundant in the lagoonal stations (∼54%) and Deltaproteobacteria in the Arachthos delta (∼49%), which was the station with the lowest oxygen concentration in the water overlaying the sediment (0.24 mg/l; Table S6).

Diversity measured as Shannon-Wiener, Pielou's evenness and Margalef's species richness indices (Table 1) was significantly different between the locations and habitats
| | OTUs | N | H′(ln) | J′ | d | Chao-1 | ACE |
|---|---|---|---|---|---|---|---|
| R_AR_A | 2,414 | 51,617 | 6.64 | 0.85 | 222.36 | 2,771.31 | 2,736.48 |
| R_AR_B | 3,070 | 74,057 | 7 | 0.87 | 273.71 | 3,450.95 | 3,353.58 |
| R_AR_C | 2,810 | 87,640 | 6.77 | 0.85 | 246.82 | 3,145.4 | 3,109.94 |
| R_ARO_A | 2,271 | 49,793 | 6.66 | 0.86 | 209.88 | 2,643.01 | 2,570.08 |
| R_ARO_B | 2,419 | 77,520 | 6.66 | 0.86 | 214.78 | 2,625.88 | 2,582.98 |
| R_ARO_C | 2,480 | 74,292 | 6.59 | 0.84 | 221.03 | 2,773.28 | 2,717.73 |
| R_ARDelta_A | 2,098 | 42,365 | 6.66 | 0.87 | 196.83 | 2,388.94 | 2,319.51 |
| R_ARDelta_B | 2,479 | 49,442 | 6.61 | 0.85 | 229.26 | 2,821.49 | 2,737.81 |
| R_ARDelta_C | 2,247 | 38,209 | 6.57 | 0.85 | 212.87 | 2,614.22 | 2,565.73 |
| L_LOin_A | 2,510 | 70,514 | 6.39 | 0.82 | 224.75 | 2,849.83 | 2,770.58 |
| L_LOin_B | 2,082 | 64,400 | 5.9 | 0.77 | 187.94 | 2,445.6 | 2,399.07 |
| L_LOin_C | 2,237 | 57,953 | 6.16 | 0.8 | 203.88 | 2,567.4 | 2,532.17 |
| L_LOout_A | 1,850 | 60,250 | 5.66 | 0.75 | 168 | 2,271.05 | 2,200.13 |
| L_LOout_B | 2,487 | 80,942 | 6.38 | 0.82 | 219.97 | 2,841.38 | 2,766.75 |
| L_LOout_C | 2,088 | 83,935 | 5.95 | 0.78 | 184.07 | 2,365.67 | 2,323.44 |
| S_Kal_A | 1,732 | 52,975 | 5.9 | 0.79 | 159.13 | 1,906.93 | 1,903.36 |
| S_Kal_B | 2,004 | 48,280 | 6.40 | 0.84 | 185.72 | 2,278.12 | 2,237.63 |
| S_Kal_C | 1,974 | 51,129 | 6.34 | 0.84 | 181.98 | 2,211.46 | 2,182.35 |

**Table 1  Diversity indices of the samples.**

**Notes.**

OTUs, total number of OTUs; N, total microbial relative abundance values; H′, Shannon-Wiener; J′, Pielou's evenness; d, Margalef's species richness; ACE, Abundance Coverage Estimator; R, River; L, Lagoon; S, Sea; AR, Arachthos; ARO, Arachthos Neochori; ARDelta, Arachthos Delta; LOin, Logarou station inside the lagoon; LOout, Logarou station in the channel connecting the lagoon to the gulf; Kal, Kalamitsi; A, B, C, replicate samples.

(Kruskal–Wallis test, Table S2). In addition, the number of OTUs, Chao-1 and ACE were also significantly different between the different habitats (Kruskal–Wallis test, Table S2). The results of the values of the Mann–Whitney $U$ tests for the *post-hoc* comparisons (Table S3) show that the main driver for the difference between the habitats is the difference of the riverine and the marine environment and secondly the difference of the former with the lagoonal environment.

## Functional community composition

The retrieved KEGG metabolic profiles, and their abundance in each sample, are provided in Table S4. For certain microbial OTUs a KEGG profile could not be retrieved, thus these OTUs constitute the fraction of unexplained taxonomic units (FTU) (*Aßhauer et al., 2015*), i.e., the amount of sequences assigned to a taxonomic unit and not transferable to KEGG reference organisms. As shown in Fig. S4 and Table S5, this fraction was highest in the lagoonal samples (67.51%) and lowest in the marine samples (38.91%). From the riverine samples (56.85%), the highest FTU was observed at the Arachthos Delta (71.32%). Due to the aforementioned FTU values, the interpretation of the results should be done cautiously as there are many OTUs for which retrieval of a metabolic profile was not possible.

However, the metabolic profiles retrieved from the OTUs were significantly different between the three habitats (Kruskal–Wallis, $p < 0.05$). Specifically, as shown in Fig. 4, there were 13 metabolic pathways that were responsible for the between-habitat dissimilarity.

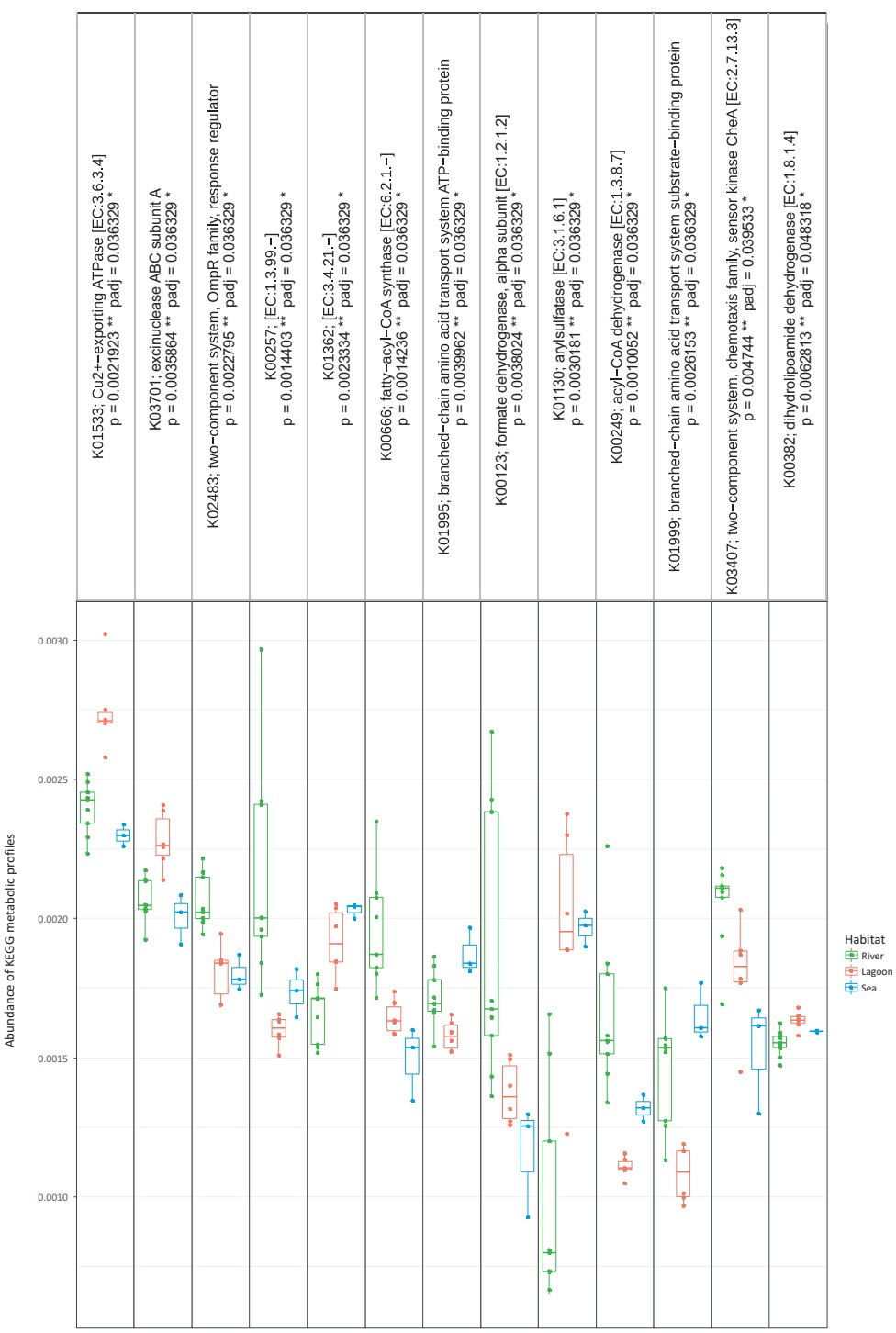

**Figure 4  The metabolic pathways that were significantly different between the three habitats.**

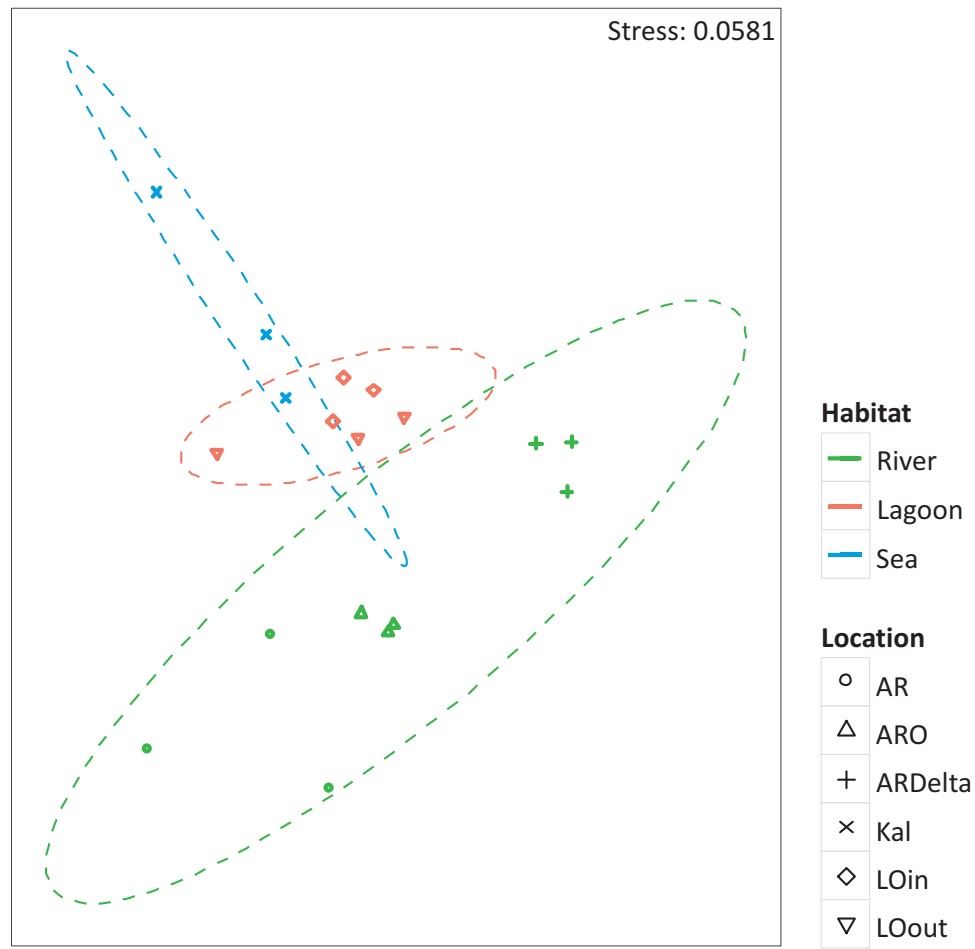

**Figure 5 nMDS of the KEGG metabolic profiles.** nMDS of the similarity matrix of the sampling stations based on the abundances of KEGG metabolic profiles. Ellipses according to habitat, signs according to location. AR, Arachthos; ARO, Arachthos Neochori; ARDelta, Arachthos Delta; LOin, Logarou station inside the lagoon; LOout, Logarou station in the channel connecting the lagoon to the gulf; Kal, Kalamitsi.

In the nMDS of the KEGG metabolic profiles (Fig. 5), it is depicted that the different habitats were functionally distinctive. This was supported by the PERMANOVA results (habitat: F.Model = 10.743, $p < 0.01$; location: F.Model = 11.206, $p < 0.01$).

## Correlation with abiotic parameters

The physicochemical variables of the sampling stations are given in Table S6. According to the results shown on Table 2, the abiotic variable that was best correlated with the microbial community pattern is salinity ($\rho = 0.88$). When the metabolic function pattern is considered, oxygen concentration was the variable showing the highest correlation with the former ($\rho = 0.73$), although the combination of salinity and oxygen concentration was also highly correlated with the metabolic pattern ($\rho = 0.63$). In addition, as shown by the variation partitioning analysis for all the combinations physicochemical variables of that resulted in models with adjusted $R^2$ of residuals less than 0.60 (Table 3), the combination of salinity, POC and temperature explained 56% of the total variation in the

**Table 2  The environmental variables best correlated with the community and the metabolic function pattern.**

| | Water | | | | Sediment | | | | | ρ |
|---|---|---|---|---|---|---|---|---|---|---|
| | Salinity (psu) | O₂ (mg/l) | Chl-a | Temperature (°C) | POC (ug/g) | Sand (%) | Silt & Clay (%) | Phaeopigments (ug/g) | CPE (ug/g) | |
| Community pattern | + | | | | | | | | | 0.88 |
| | + | | | | + | | | | | 0.85 |
| | + | + | | | + | | | | | 0.79 |
| | + | + | | | + | | + | | | 0.72 |
| | + | + | | | + | | + | | + | 0.66 |
| | + | + | | + | + | | + | + | | 0.64 |
| | + | + | | + | + | + | + | + | | 0.58 |
| | + | + | + | + | + | + | + | + | | 0.54 |
| Metabolic function pattern | | + | | | | | | | | 0.73 |
| | + | + | | | | | | | | 0.63 |
| | | + | | + | | | + | | | 0.57 |
| | + | + | | | | + | + | | | 0.54 |

**Notes.**

POC, Particulate Organic Carbon; CPE, Chloroplastic pigment equivalents; Chl-a, Chlorophyll-a concentration (ug/l) and Fluorescence; ρ, Spearman rank correlation coefficient.

community similarity matrix. When the explanatory variables were regarded separately, salinity accounted for 27% of the variation, followed by POC (21%) and temperature (14%). In metabolic function pattern, 50% of the variation was explained by salinity and oxygen concentration; salinity alone accounted for 12% of the variation while oxygen concentration for 33%.

Regarding the relationship between the number of OTUs and the salinity values, as shown in Fig. 6, a linear decrease of the former was observed from the freshwater to the marine stations; salinity explained over 65% of the variation in the number of OTUs. Robust models were also evident when the data were divided into taxonomic groups (Table S7), with salinity explaining from 70% up to 98% of the variation of the OTUs diversity, in the case of Firmicutes and TM7 respectively. The residuals of all significant models presented in Table S7 showed no evidence of heteroscedasticity and were found to be normally distributed.

## DISCUSSION

### Microbial community composition

Based on the results of the present study, it is suggested that the microbial community diversity pattern differs by habitat and location, thus indicating that, at least to some extent, each habitat hosts a different microbial community from the others, regarding both the number of OTUs and their composition, as has been also shown from similar studies in the Baltic Sea (Herlemann et al., 2011). Furthermore, the abiotic variable that is best correlated with the microbial community pattern is salinity, as it has been shown from studies on bacterioplankton (e.g., Kirchman et al., 2005; Nemergut et al., 2011; Fortunato et al., 2012),

**Table 3** The percentage of variation explained (adjusted $R^2$) of each explanatory physicochemical variable, as well as their combinations.

| | Community pattern | Metabolic function pattern |
|---|---|---|
| Salinity | 27%[***] | 12%[*] |
| $O_2$ | | 33%[**] |
| Temperature | 14%[**] | |
| POC | 21%[***] | |
| Salinity + $O_2$ | | 50%[***] (residuals: 50%) |
| Salinity + Temperature | 47%[***] (residuals: 53%) | |
| Salinity + POC | 45%[***] (residuals: 55%) | |
| POC + Temperature | 36%[***] | |
| Salinity + POC + Temperature | 56%[***] (residuals: 44%) | |
| Salinity with Temperature as condition variable | 32%[***] | |
| Salinity with POC as condition variable | 24%[***] | |
| Salinity with $O_2$ as condition variable | | 16%[**] |
| Salinity with Temperature and POC as condition variables | 20%[***] | |
| Temperature with Salinity as condition variable | 20%[***] | |
| Temperature with POC as condition variable | 15%[***] | |
| Temperature with Salinity and POC as condition variables | 11%[**] | |
| POC with Salinity as condition variable | 18%[***] | |
| POC with Temperature as condition variable | 22%[***] | |
| POC with Salinity and Temperature as condition variables | 10%[**] | |
| $O_2$ with Salinity as condition variable | | 37%[***] |

**Notes.**
POC, Particulate Organic Carbon.
[*]$p < 0.05$.
[**]$p < 0.01$.
[***]$p < 0.001$.

as well as sediment bacterial communities (*Bolhuis & Stal, 2011*; *Severin, Confurius-Guns & Stal, 2012*; *Bolhuis, Fillinger & Stal, 2013*; *Pavloudi et al., 2016*).

The majority of the observed OTUs was classified as Bacteroidetes and Proteobacteria. Bacteroidetes have been found to be omnipresent along estuarine gradients (e.g., *Bouvier & Del Giorgio, 2002*), although in certain cases they dominated higher salinity communities (*Campbell & Kirchman, 2013*; *Dupont et al., 2014*) which has been attributed to their ability for degradation of complex organic matter (*Blümel, Süling & Imhoff, 2007*). In the present study, their abundance indeed increased in the brackish stations but decreased in the marine station, which could be due to the lower availability of organic matter in the latter, as reflected in the concentration of particulate organic carbon.

Alphaproteobacteria dominate marine communities (*Edmonds et al., 2009*; *Campbell & Kirchman, 2013*; *Herlemann et al., 2016*) but they are also ubiquitous in freshwater

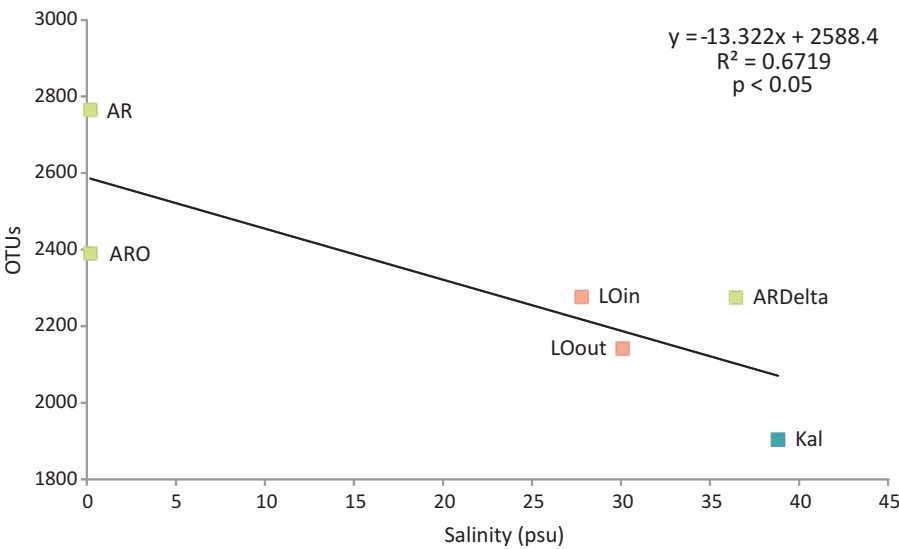

**Figure 6** **Linear regression between the number of OTUs (averaged per sampling station) and the salinity of the sampling stations.** Color according to habitat. AR, Arachthos; ARO, Arachthos Neochori; ARDelta, Arachthos Delta; LOin, Logarou station inside the lagoon; LOout, Logarou station in the channel connecting the lagoon to the gulf; Kal, Kalamitsi.

habitats (*Zhang et al., 2014*), which justifies their high abundance in both the mouth of the Arachthos river and the Kalamitsi stations. Betaproteobacteria were almost exclusively present in the inner station and mouth of the Arachthos river; this complies with the general trend of Betaproteobacteria dominating freshwater habitats (*Campbell & Kirchman, 2013*; *Zhang et al., 2014*) and declining with increasing salinity (*Wu et al., 2006*). Their absence from the other sampling stations can be explained by the fact that they are typical freshwater bacteria (e.g., *De Bie et al., 2001*), adapted to live in low salt concentrations and low osmotic pressure (*Zhang et al., 2014*).

Gammaproteobacteria are generally more abundant in higher salinities (*Wu et al., 2006*; *Zhang et al., 2014*; *Herlemann et al., 2016*), which has been attributed to their opportunistic life strategies (*Pinhassi & Berman, 2003*) and the low salinity conditions being unfavorable for their growth (*Zhang et al., 2014*). However, Gammaproteobacteria can dominate brackish habitats (*Edmonds et al., 2009*) as has also been shown from previous studies in the same sampling stations (*Pavloudi et al., 2016*) and from the results of the present study.

Deltaproteobacteria were mostly abundant in the Arachthos delta; this can be attributed to the hypoxic conditions prevailing in this sampling station (*Crump et al., 2007*) and to their generally documented abundance in brackish habitats (*Edmonds et al., 2009*; *Pavloudi et al., 2016*).

The low abundance of Archaea could be attributed to the primers used for the present study, which were not specific for amplification of the Archaeal communities. In addition, the low abundances found can be traced to their inability to grow under estuarine environmental conditions since they are primarily of allochthonous origin (*Bouvier & Del Giorgio, 2002*). However, the abundances of Archaea were decreasing with increasing salinity; in the marine environment, Archaea are generally limited to shallow or deep-sea

anaerobic sediments and extreme environments (*DeLong, 1992*), which could explain their absence from the marine station in the present study.

It is evident that sea level rise, and the subsequent saltwater intrusion that it will cause, will impose an osmotic stress in microbial populations in the riverine stations (*Chambers, Reddy & Osborne, 2011*). This may eventually shift the community in a more brackish state, although this will depend on the time-scale and intensity of saltwater intrusion. Microbial communities in fluctuating environments have been shown to be rather resilient, which causes variations in their expected functional response to change (*Hawkes & Keitt, 2015*). However, saltwater intrusion may induce an indirect effect in microbial communities, for example by causing changes in vegetation (*Nelson et al., 2015*) or by increasing sulfate concentrations, thus by stimulating its reduction (*Chambers, Reddy & Osborne, 2011*).

As far as the relationship between the number of OTUs and salinity values is concerned, it has been shown that it is negative, i.e., diversity decreases with the increase of salinity. Thus, the sediment microbial communities in the Amvrakikos Gulf salinity gradient do not appear to follow Remane's concept but rather the linear model proposed by *Attrill (2002)* with the species minimum at the point of maximum salinity range. This is in contrast with the constant relationship observed by *Herlemann et al. (2011)* for the Baltic Sea bacterioplankton along the salinity gradient. The divergence from the species-minimum concept could be attributed to the different life strategies of micro- and macroorganisms, as has been suggested by *Telesh, Schubert & Skarlato (2013)*. In addition, microorganisms can be transported with water movement, apart from experiencing adaptation only at the molecular and cellular level (*Telesh, Schubert & Skarlato, 2015*). Thus, they experience salinity stress diffently from benthic animals with reduced mobility (*Skarlato & Telesh, 2017*), which causes their deviation from the recognized models for macrobenthic organisms.

Although a decreasing trend of microbial diversity along gradients of increasing salinity has been observed (*Rodriguez-Valera et al., 1985*; *Benlloch et al., 2002*), the results of our study showed that this was only evident for the total number of OTUs; the other diversity indices did not show a statistically significant negative relationship with salinity (data not shown).

## Functional community composition

The higher number of FTU observed in the lagoonal samples is indicative of the high microbial community diversity of this habitat. It also reflects the need for more extensive studies in order for an enhanced taxonomic resolution to be achieved for sequences found in lagoons. Similarly, the lower FTU was found in the marine samples, thus reflecting the number of studies that have been conducted in the marine environment so far which allowed for more OTUs to be transferred to KEGG reference organisms. Analogous studies in Chesapeake Bay, which is also a large water body with a pronounced salinity gradient, have also suggested that protein identification in environmental samples is rather challenging. Sequence databases have derived from cultured organisms and thus, it is unlikely that significant matches can be found between the bacterioplankton metaproteome and the queried databases (*Kan et al., 2005*). Hence, more studies on isolation and cultivation of

microorganisms from these habitats are needed in order to enrich the available information on KEGG and similar databases.

Although retrieval of a metabolic profile was not possible for the majority of the OTUs in certain cases, it is shown that different habitats were functionally distinctive and that salinity and oxygen concentration were highly correlated with the retrieved metabolic pattern. This is in accordance with studies suggesting that the actual patterns of composition and metabolism transition are strongly linked to hydrological conditions (*Bouvier & Del Giorgio, 2002*). Furthermore, salinity has been found to be the main factor explaining almost all differences in the key metabolic capabilities of the Baltic Sea bacterial communities (*Dupont et al., 2014*), so a similar relationship could be expected for the Mediterranean analogue of the Baltic Sea. In addition, salinity has been shown to influence proteome profiles from bacterioplankton communities of the Chesapeake Bay, since samples with higher salinity, i.e., closer to marine origin, are more similar and, at the same time distinct from, inner Bay samples (*Kan et al., 2005*). Similar results have been found for the San Francisco Bay, where salinity is one of the key variables influencing the abundance and activity of ammonia-oxidizing prokaryotes and denitrifiers (*Mosier & Francis, 2008*; *Mosier & Francis, 2010*).

## CONCLUSIONS

From the results of the present study, it can be concluded that the sediment microbial OTUs of the Amvrakikos Gulf salinity gradient do not follow the Remane's concept, i.e., there is no decrease in the intermediate salinities, but rather a negative trend. In addition, different taxonomic groups were more abundant in the freshwater stations while others were more abundant in the marine environment. Salinity was also found to influence the metabolic function patterns that were retrieved for the sampling stations. However, future studies are needed to decipher the metabolic capabilities of all the OTUs found at the habitats under study and investigate in depth the impact of salinity at their functional potential. Furthermore, experimental studies are needed in order to directly examine the effect of salinity on microbial community composition and investigate how the latter will respond when subjected to salinities varying from freshwater to marine in the light of climate change and sea level rise.

## ACKNOWLEDGEMENTS

The authors would like to thank the Amvrakikos Wetlands Management Body (http://www.amvrakikos.eu/), and especially Mr Spyridon Konstas and Mr Demetrios Barelos, for providing access to the studied lagoons and for their valuable support during our sampling campaign. In addition, the authors would like to thank the captain and crew of the R/V Philia and Dr. Manolis Tsapakis for providing the samples and CTD data from the Arachthos delta and the Kalamitsi station. Also, the authors would like to thank the members of the Biodiversity lab of HCMR, and specifically Dr. Georgios Chatzigeorgiou, Dr. Eva Chatzinikolaou, Ms. Kleoniki Keklikoglou and Dr. Katerina Vasileiadou for their valuable assistance during the sampling campaign.

Furthermore, the authors would like to thank Dr. Umer Zeeshan Ijaz for providing access to the Orion IT cluster, where the analyses were performed, and for his bioinformatics tutorial entitled "Illumina Amplicons OTU Construction with Noise Removal" (available at http://www.tinyurl.com/JCBioinformatics). This work is part of a MARES Doctoral Research (MARES_12_13). MARES is a Joint Doctorate programme selected under Erasmus Mundus coordinated by Ghent University (FPA 2011-0016) (http://www.mares-eu.org/).

### Funding

This work was supported by the LifeWatchGreece (European Strategy Forum on Research Infrastructures) project (http://www.lifewatchgreece.eu/) [384676-94/GSRT/NSRF(C&E)] and the EU BON (Building the European Biodiversity Observation Network) project (http://www.eubon.eu/), funded by the European Union under the 7th Framework programme, Contract No. 308454. There was no additional external funding received for this study. The funders had no role in study design, data collection and analysis, decision to publish, or preparation of the manuscript.

### Grant Disclosures

The following grant information was disclosed by the authors:
LifeWatchGreece (European Strategy Forum on Research Infrastructures) project: 384676-94/GSRT/NSRF(C&E).
EU BON (Building the European Biodiversity Observation Network) project.
European Union under the 7th Framework programme: 308454.

### Competing Interests

The authors declare there are no competing interests.

### Author Contributions

- Christina Pavloudi conceived and designed the experiments, performed the experiments, analyzed the data, contributed reagents/materials/analysis tools, wrote the paper, prepared figures and/or tables, reviewed drafts of the paper.
- Jon B. Kristoffersen and Anastasis Oulas contributed reagents/materials/analysis tools, reviewed drafts of the paper.
- Marleen De Troch reviewed drafts of the paper.
- Christos Arvanitidis conceived and designed the experiments, reviewed drafts of the paper.

### Field Study Permissions

The following information was supplied relating to field study approvals (i.e., approving body and any reference numbers):

The permission to conduct the field study was provided by the Amvrakikos Wetlands Management Body (http://www.amvrakikos.eu/).

## DNA Deposition

The following information was supplied regarding the deposition of DNA sequences:

All the raw sequence files of this study were submitted to the European Nucleotide Archive (ENA) with the study accession number PRJEB20211 (Available at http://www.ebi.ac.uk/ena/data/view/PRJEB20211).

## Data Availability

The raw data has been supplied in the Tables S1–S7.

## Supplemental Information

Supplemental information for this article can be found online at http://dx.doi.org/10.7717/peerj.3687#supplemental-information.

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
