# Peer review of "Sediment microbial taxonomic and functional diversity in a natural salinity gradient challenge Remane’s “species minimum” concept"

_PeerJ, doi:10.7717/peerj.3687_

## Round 0.1 · original submission · Major Revisions

All three reviewers are in agreement that your research is significant and worthy of publication. However, they have also expressed concerns about the cohesiveness and focus of the manuscript. The title and abstract should clearly represent the question addressed--whether a test of Remane's concept or a broader study of the sediment microbial community--and the relevance of each method used needs to be clear.

I elected for "Major Revisions" rather than "Minor" so that the reviewers will see the revised manuscript, which I believe is important.

Please also provide a point-by-point response to the reviewers' reports, detailing where and how you have revised your manuscript.

I look forward to receiving your revised manuscript.

·

Basic reporting

The authors test the suitability of a macrobenthic ecological observation due to salinity gradients, on sediment microbial community structure and function. The authors provide sufficient background on the original ecological concept, their study system, and their methods. Raw data are available. Data and figure reformatting is suggested. Minor grammar and word changes in the manuscript are recommended.

Experimental design

The authors use 16S rRNA gene sequencing of sediments from a fresh-to-marine transect, and typical metrics to analyze the role of habitat and environmental influences on microbial communities. The study seems to build off a previous publication from the same group, that evaluates the influence of salinity on bacterial community structure in lagoons in the same sampling location (http://www.sciencedirect.com.proxy.lib.umich.edu/science/article/pii/S1874778716300058), but the current study incorporates a broader salinity range than previously evaluated. The authors infer functional characteristics of observed bacterial communities, but functions for 38-75% of taxa in each sample cannot be obtained. The authors also tentatively evaluate archaeal community composition, of which this study design is not optimal.

Validity of the findings

The authors conclude that Remane’s “species minimum” concept does not translate to sediment microbial community structure and function. Their main supports are 1). A decrease in species richness over the salinity gradient; and 2). Strong correlation between salinity and community composition, and salinity and function. The authors rightly suggest caution when interpreting archaeal relative abundances and functional profiles from bacterial 16S rRNA gene data. The authors use a paragraph in the introduction discussing the unknown impacts of sea level rise and salinity changes on freshwater systems. Their study system could represent a microbial community in transition from fresh to saline conditions, and functional signatures could be evaluated in that context; the authors could address this compelling ecological consequence in their discussion.

Additional comments

Please review paragraph breaks for clarity and cohesion. There are examples of unexpected paragraph breaks at lines 114-115, the methods section for “DNA extraction, PCR amplification and 16S rRNA sequencing”, 340-341, and 369-370.
Please check grammar and flow. For example, lines 224-225 could be clearer: “The number of sequences assigned to an OTU in each sample represents the relative abundance of the OTU in a sample.” Suggestion for lines 201-203: “The raw sequence reads retrieved from all the sediment samples were quality trimmed using sickle (Joshi & Fass, 2011), to where the average quality score dropped below 20 (-q 20) as well as where read length was below 10bp (-l 10).”
Methods: Lines 133-139 detail two different instruments to measure dissolved oxygen, salinity, and temperature. Were these instruments cross-calibrated? Please see comment about Supplementary Table 6.
Lines 169-172: This statement implies that the authors used multiple primer sets on different samples, and compared resulting community profiles. However, primers impart community biases, and the use of different primer sets in the same study is puzzling. Please provide background research on or results comparing bias from the selected primer pairs.
Lines 224-226: Are your analyses on each station using averaged representations of OTUs across the replicates? If so, do you have tests on the similarity of the replicates?
Figure 2: Are there any significance values attributed to the ellipses, or are they just encircling the station points?
Figures 3, S2, and S3: Please clarify if each bar represents the relative abundance of phyla in one sample at each station, or if the relative abundance profiles of samples at each station were averaged to get a mean representation of phyla at each station. If they were averaged, see above comment about methods. Graph axes need labels.
Figure 4: Vertical lines separating each triplet per KEGG would make interpreting the graph easier. Axes need labels.
Figure 6: Would error bars on OTU abundances be visible on this graph, or be obscured by the station points?
Supplementary Table 1: Please change “paired-end reads” to “read pairs”. One overlapped read is the result of two paired-end reads.
Supplementary Table 6: Chlorophyll a is reported for all stations, however the methods state that fluorescence ~ chlorophyll a was measured at only Arachthos delta and Kalamitsi stations. How was fluorescence or chlorophyll a measured at all the other stations? Also, the chlorophyll a values for those latter locations have a different format than the rest.
Discussion: Your study has a clean and simple design, of using the sediment microbial communities and relevant environmental parameters to test Remane’s species minimum in other systems. I think the results will have broader significance if you can relate them to benthic ecological succession due to rising sea level and climate change.

·

Basic reporting

no comment.

Experimental design

no comment.

Validity of the findings

-Data is robust, but statistical analysis could be improved to specifically test the role of salinity in driving community patterns.
-Conclusion are well stated, but not linked to original research question as novel aspects (community variation and functional profiles) were added and not justified at the first place.

Additional comments

I praise the authors for their extensive data set and for their thorough statistical analyses. The manuscript is clearly written and there are only minor points to be checked for improved clarity. Yet, some points should be addressed to make the study stronger as follows:

1) The study would gain from being more focused on what the title and abstract described, i.e. the testing of the theory that "taxonomic diversity of microbenthic organisms is lowest within the horohalinicum". The manuscript should have thus consisted of testing local microbial richness (alpha diversity) as a function of the salinity gradient. The authors extended their analysis to community turnover and to the functional relevance of their observed diversity patterns, which in my opinion, analyses that dilute the main message of the study or at least have not been justified with respect to Remane’s theory.
I would suggest that the authors either change their title and abstract to be broader in scope, or fit their text content to the testing of Remane’s theory as main objective.
2) Some technical points are not clear:
2a. Line 213: why did the reads have to be mapped back to the OTUs? AT this stage of the procedure, all reads should be clustered into OTUs. Please clarify.
2b. Line 209 the singletons were removed, and few lines later (line 213), they were added back. Is this a normal procedure (please provide references then)? It seems that this procedure may add sequences that will not fit to any OTUs because those reads were not considered in building the OTU matrix at the first place.
3) One main point that the authors touched upon is whether salinity is the main driver of community patterns or not. The current analysis based on correlations does not account for factor confounding and for the amount of variation each factor might explain. To address this point, I suggest that the authors perform a variation partitioning analysis (see vegan package varpart function for instance) in order to quantify the amount of variation in community that is due to salinity uniquely while taking other environmental and spatial factors into consideration. Hence, factor confounding on community variation can be disentangled. The same approach can be applied to changes in richness and would be a strong point to be made in this study.
4) The salinity gradient (e.g. Figure 6) does not include samples at the 5-8 psu that are mentioned in Remane’s concept. This would suggest that the authors should reinforce the idea that the authors should refocus their study on the effect of salinity and biogeography on community variation, instead of addressing Remane’s concept.
Minor comments
1) Line 197: The reference Leinonen et al., 2011 could be removed or put after "ENA" on line 196.
2) Figure 2. The legend "nMDS of the microbial OTUs" is not a valid legend, as nMDS applies to a distance matrix. Please rephrase.
3) Lines 325-326: please rephrase as the sentence is intrinsically redundant.

·

Basic reporting

No comment

Experimental design

No comment

Validity of the findings

No comment

Additional comments

General comments:

1) Lines 41-42:
In the Abstract, the authors provide an important statement that “the sediment microbial OTUs in this study area do not follow Remane’s concept which could be attributed to the different life strategies of micro- and macroorganisms”. Although the idea of differences in life strategies between unicellular plankton and large multicellular bottom-dwelling organisms is highly relevant to the results of this study, it is, unfortunately, neither mentioned in the Introduction nor discussed elsewhere in the manuscript. Moreover, this idea was originally published by Telesh et al. (2013) and then referred to in several other articles, e.g. by Herlemann et al. (2016), who wrote: “Telesh et al. (2013) suggested that the life strategies of unicellular planktonic organisms differ substantially from those of large multicellular bottom-dwelling organisms, resulting in deviations from the species-minimum concept, which is supported by the results from our study of bacterioplankton” (Herlemann et al., 2016, page 7). Therefore, I definitely recommend considering this viewpoint in the manuscript and discussing it, providing the appropriate citation(s). Alternatively, this statement should be deleted from the Abstract since it is not a result obtained by the authors of this submission.

2) Lines 101-103:
It is very difficult to agree with the authors’ statement that “…confirmation of Remane’s concept for prokaryotes would mean that small-bodied, fast-developing and rapidly evolving microbes respond in a similar way as benthic macroorganisms in a salinity gradient”, mainly for the following reasons.
First, this is likely not so because of specific morphological, physiological and ecological characteristics of these groups of organisms. Thus, the differences in their reaction to salinity gradient will always be present due to dissimilarities in size, mode of life, osmoregulation etc. These issues are discussed in many details in Telesh et al. (2011, 2013, 2015 and 2016).
Second, even if diversity of sediment prokaryotes happen to follow the Remane’s curve, one should look for more appropriate (maybe even unknown yet) reasons for this pattern. In this respect, a good example was provided in the paper by Schubert et al. (2011, Mar. Pollut. Bull., Vol. 62) where the authors revealed the Remane’s curve-like pattern in the diversity of macrophytes in the salinity gradient in certain cases; however, the reasons for the species minimum of those macroalgae were the lack of suitable hard substrates but not at all the critical salinity (horohalinicum) in those habitats.
Therefore, I suggest that the authors provide deeper analysis/explanations of the obtained results and a broader discussion of the possible mechanisms that underpin those findings.

3) Lines 313-321:
The result which showed that for certain microbial OTUs a KEGG profile could not be retrieved, thus these OTUs constituted the fraction of unexplained taxonomic units (FTU), and that this fraction was the highest in the lagoon samples (67.51 %) and lowest in the marine samples (38.91 %) is of special interest. I agree with the authors that this issue needs further investigation which might shed more light on the reasons and mechanisms behind the natural prokaryote diversity distribution pattern in the salinity gradient. However, I believe that the existing knowledge on microbial diversity in the other large water bodies with salinity gradient (e.g., the San Francisco Bay, or the Chesapeake Bay and its sub-estuaries) allows for certain generalizations already now. The paper will benefit from the enlarged discussion of these important issues.

Specific suggestions:

1) Line 69: peak (not “pick”).
2) Lines 280-281 and elsewhere in the text: I suggest to change “it seems” to “it is likely” or “we believe that” etc.
3) Legend to Figure 4:
Please consider salinity gradient while compiling the legend and change the order of symbols in Fig. 4 to follow the gradient: river -> lagoon -> sea.

---

## Round 0.2 · accepted · Accept

Two of the original reviewers did not comment on the revised manuscript, but the third approved the changes and recommended acceptance.

·

Basic reporting

No comment

Experimental design

No comment

Validity of the findings

No comment

Additional comments

I accept all authors' responses to my critical comments and suggestions.

One minor 'technical' correction should be made: in Line 725 of the MS, the format is corrupted (two references are not separated by the line break).